# Analysis of Spatial–Temporal Variability of PM$_{2.5}$ Concentrations Using Optical Satellite Images and Geographic Information System

**Dewinta Heriza** [1] , **Chih-Da Wu** [1] , **Muhammad Aldila Syariz** [2] **and Chao-Hung Lin** [1,*]

1   Department of Geomatics, National Cheng Kung University, Tainan City 70101, Taiwan;
    p68097080@gs.ncku.edu.tw (D.H.); chidawu@mail.ncku.edu.tw (C.-D.W.)
2   Department of Geomatics Engineering, Institut Teknologi Sepuluh Nopember, Surabaya 60111, Indonesia;
    aldilasyariz@its.ac.id
*   Correspondence: linhung@mail.ncku.edu.tw

**Abstract:** Particulate matter less than 2.5 microns in diameter (PM$_{2.5}$) is an air pollutant that has become a major environmental concern for governments around the world. Management and control require air quality monitoring and prediction. However, previous studies did not fully utilize the spectral information in multispectral satellite images and land use data in geographic datasets. To alleviate these problems, this study proposes the extraction of land use information not only from geographic inventory but also from satellite images with a machine learning-based classification. In this manner, near up-to-date land use data and spectral information from satellite images can be utilized, and the integration of geographic and remote sensing datasets boosts the accuracy of PM$_{2.5}$ concentration modeling. In the experiments, Landsat-8 imagery with a 30-m spatial resolution was used, and cloud-free image generation was performed prior to the land cover classification. The proposed method, which uses predictors from geographic and multispectral satellite datasets in modeling, was compared with an approach which utilizes geographic and remote sensing datasets, respectively. Quantitative assessments showed that the proposed method and the developed model, with a performance of RMSE = 3.06 μg/m$^3$ and R$^2$ = 0.85 comparatively outperform the models with a performance of RMSE = 3.14 μg/m$^3$ and R$^2$ = 0.68 for remote sensing datasets and a performance of RMSE = 3.47 μg/m$^3$ and R$^2$ = 0.79 for geographic datasets.

**Keywords:** fine particulate matter; land use regression; optical satellite images

## 1. Introduction

Fine particulate matter (PM$_{2.5}$) consists of solid or liquid particles with tiny diameters (2.5 micrometers) suspended in atmospheric gases. The particles that contain metal and organic components may induce free radicals to produce oxidized lung cells, which is the main cause of respiratory system injury [1]. The respiratory injury may affect human health in the short term (e.g., chronic obstructive pulmonary disease and acute nasopharyngitis) and in long-term period (e.g., cardiovascular disease and mortality) [2–4]. In the worst cases, in which the exposure of PM$_{2.5}$ is high [5–7], the particles threaten not only adults but also children, because of the immature state of their lungs and immune function [8]. This instance implies that exposure to PM$_{2.5}$ is detrimental to human health; therefore, the monitoring of PM$_{2.5}$ concentrations is critical. Governments generally install stationary air quality sensors to measure PM$_{2.5}$ concentrations and other air quality parameters, including PM$_{10}$, SO$_2$, O$_3$, CO, NO$_2$. The air quality-monitoring stations are only distributed and located in specific interesting landscapes, such as the middle of towns, industrial areas, and forests. Thus, methods to derive PM$_{2.5}$ concentrations for non-stationary regions are required for the monitoring of an entire region.

Land use regression (LUR), which predicts air quality parameters of interest from land use geographic datasets, was proposed [9], and the method can be adopted to obtain $PM_{2.5}$ concentrations at non-stationary locations [10–12]. An LUR model for $PM_{2.5}$ concentration was created by Lee et al., with several traffic-based predictors such as traffic intensity, road length, and road proximity [13]. The Eeftens study included population-based data for six classes of predictors, including high-density residential, low-density residential, industry, ports, urban green, and natural land [14]. Liu et al. and Yu et al. utilized culture-specific sources of $PM_{2.5}$, such as joss paper burning at temples and cooking smoke from restaurants in East Asian countries, respectively [15,16]. Although these studies achieved high-accuracy prediction, the estimation of $PM_{2.5}$ concentrations is hindered by the slow updating of land use geographic datasets. The updating of land use data is generally time-consuming and labor sensitive. The time cycle for land use data updating takes years, which may cause biases in the modeling of $PM_{2.5}$ concentrations. Tunno's study extended the predictors by using other chemical particles, including ammonia, carbon monoxide, and sulfur dioxide [17], based on the fact that $PM_{2.5}$ exposure is explained as the secondary product of the aforementioned particles, the data of which can be obtained in air quality-monitoring stations [18,19]. To assess the non-stationary points, an interpolation technique, such as inverse distance weighting (IDW) or kriging, is performed [20–22].

Passive remote sensing satellite sensors record the reflected sunlight radiances in varying wavelengths, including visible and infrared spectral ranges, from the Earth's ground surface. The satellite revisit time is around 2–20 days, depending on the design of the satellite orbit and constellation. The variation in radiances in spectral bands enables the detection and classification of ground objects and the extraction of the Earth's surface features, which are useful for Earth observation and remote sensing applications. For instance, Wu et al. utilized the normalized difference vegetation index (NDVI), calculated from moderate resolution imaging spectrometer (MODIS) images with 1 km spatial resolution [23]. The NDVI was set as one of the predictors in the development of the LUR model for $PM_{2.5}$ concentration estimation in Taiwan. The NDVI, based on the red and near-infrared spectral bands which can absorb pollutants, is used to represent the density of vegetation covers and greenness [24–26]. In addition to NDVI, the satellite images can provide information on artificial covers and water bodies from multispectral bands. The revisit time of MODIS, which can increase the temporal resolution of satellite images, is two days. However, the spatial resolution of MODIS images is 500 m $\times$ 500 m, implying that small objects, including small waterbodies and residential regions, cannot be represented well in the images. Consequently, the spatial resolution of land covers in MODIS images cannot match well with that in the geospatial dataset.

The main idea of this study is to integrate the geospatial data, which contain land use, landmarks, digital road networks, and digital terrain models (DTM), and the optical satellite data, which contain NDVI and land cover classification, in the modeling of $PM_{2.5}$ concentrations. This integration can complement the advantages in the geospatial datasets and remote sensing images. The study proposes the use of a Landsat-8 operational land imager (OLI) sensor instead of MODIS because of its higher 30 m spatial resolution, which improves the classification of small objects. In addition, cloud pixels of a satellite image on a particular acquisition date are masked out and replaced with the spectral information of corresponding pixels from other acquisition dates to generate a cloud-free image. Thus, the proposed method provides the following contributions: (1) the integration of satellite images with a geographic dataset, which provides suitable temporal and spatial resolution land use information and contributes to $PM_{2.5}$ concentration estimation; and (2) the generation of cloud-free images, which enables the classification of land covers without missing regions. The remainder of this paper is organized as follows. Section 2 introduces the study area and datasets, and Section 3 proposes the methodology. Section 4 shows the experimental results, and Section 5 provides the conclusions.

## 2. Study Area and Datasets

### 2.1. Taipei Metropolis and Air Quality Dataset

Taipei Metropolis, which consists of Taipei City and New Taipei City and is located in northern Taiwan, was selected as the study area. Since the capital movement in 1884, Taipei Metropolis has been the hub of governmental, economic, and cultural activity in Taiwan, and has more than one-fourth of Taiwan's total population [22,27]. To maintain the air quality alongside the large-scale human activity, the Environmental Protection Agency (EPA) of Taiwan installed 17 automatic air quality-monitoring stations in Taipei Metropolis (Figure 1a) to understand and maintain the air's characteristics and quality. The stations are classified into three categories: general, traffic, and national park stations. The general and traffic stations are found in populous and heavy traffic areas, respectively, whereas the national park stations are installed in the national parks that are far from residential areas.

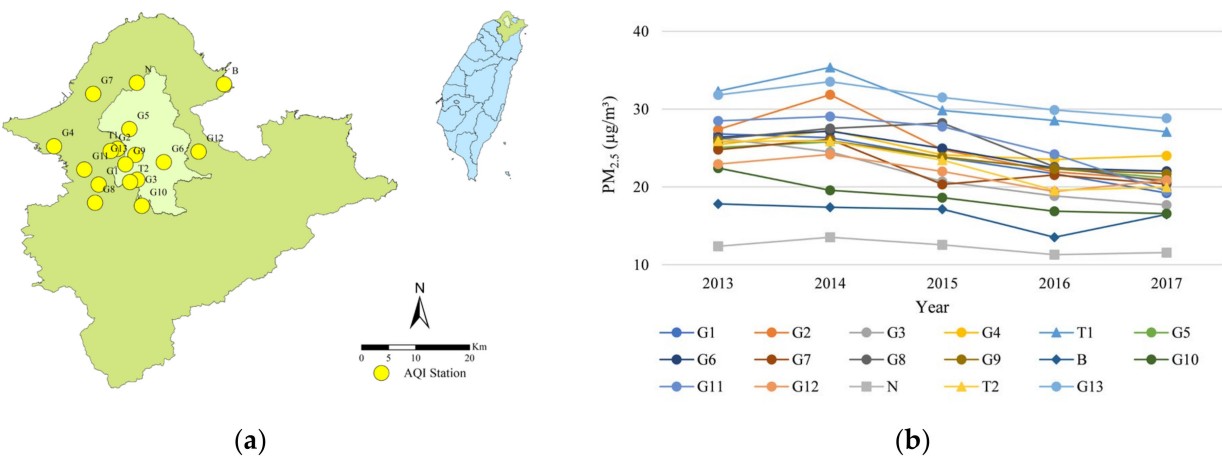

(**a**)  (**b**)

**Figure 1.** Study area and $PM_{2.5}$ monitoring stations. (**a**) Locations of the study area (Taipei Metropolis) and $PM_{2.5}$ monitoring stations. (**b**) Time series of the $PM_{2.5}$ concentrations from 2013–2017 (17-monitoring stations).

The stations record several air quality parameters, including $PM_{2.5}$ concentrations. The time series trends of the collected $PM_{2.5}$ concentrations from 2013 to 2018 are shown in Figure 1b. The $PM_{2.5}$ concentrations at the 17 stations range from 10 to 40 $\mu g/m^3$, levels which are of moderate air quality in terms of health concerns. The air quality is acceptable, but may affect people who are sensitive to air quality. In addition to $PM_{2.5}$, the stations also provide other air quality parameters, including the concentrations of ozone ($O_3$), particulate matter ($PM_{10}$), carbon monoxide (CO), sulfur dioxide ($SO_2$), and nitrogen dioxide ($NO_2$), rainfall, wind speed, and wind direction. These parameters are set as potential predictors in the development of a model for the estimation of $PM_{2.5}$ concentrations in this study. A total of 102 air quality records were collected from the 17 stations from 2013–2018. Since the air quality parameters are available only at monitoring stations, these parameters, excluding $PM_{2.5}$, in non-station regions were interpolated by means of inverse distance weighting interpolation (IDW). These interpolated data are used for the estimation of $PM_{2.5}$ concentrations at the non-station points.

### 2.2. Geographic Dataset

Five geographic datasets, including the national land use inventory, the digital road network map, landmark dataset, a map of the industrial park, and DTM, as described in Table 1, were used. The national land use inventory was generated by using aerial images from 2006 to 2008 with 1:5000 scale and stored as a polygon layer in a geographic information system [28]. The inventory includes four land use classes: agriculture, pure residential, commercial residential, and industrial–commercial residential. The last two classes refer to residential areas that contain artificial buildings for small industries and

commercial activities, respectively. The digital road network produced by the Ministry of Transportation and Communication, Taiwan, was utilized. The roads are reclassified into three different classes, that is, local roads, major roads, and expressways. The landmark dataset consists of more than 0.25 million records that refer to the locations of restaurants, temples, and others.

**Table 1.** Geographic datasets of Taipei Metropolis and related predictors used in $PM_{2.5}$ estimation.

| Geographic Datasets | Types | Predictor Notations |
|---|---|---|
| National land use inventory | Pure residential (pR), commercial residential (cR), industrial–commercial residential (iR), agricultural (A) | $\{pR_{250m}, \cdots, pR_{2000m}\}$, $\{cR_{250m}, \cdots, cR_{2000m}\}$, $\{iR_{250m}, \cdots, iR_{2000m}\}$, $\{A_{250m}, \cdots, A_{2000m}\}$ |
| Map of industrial park | Industrial parks in the year of 2010 (iP) | $\{iP_{250m}, \cdots, iP_{2000m}\}$ |
| Landmark | 0.25 million landmarks including Chinese restaurant (CR), night market (NM), temple (Te) | $\{CR_{250m}, \cdots, CR_{2000m}\}$, $\{NM_{250m}, \cdots, NM_{2000m}\}$, $\{Te_{250m}, \cdots, Te_{2000m}\}$ |
| Digital road network | Local roads (LR), major roads (MR), and expressways (EW) | $\{LR_{250m}, \cdots, LR_{2000m}\}$, $\{MR_{250m}, \cdots, MR_{2000m}\}$, $\{EW_{250m}, \cdots, EW_{2000m}\}$, |
| DTM | DTM with the 20 m spatial resolution (DTM) | |

Buffer analysis is an essential spatial analysis function in GIS. This analysis constructs a zonal area, which may include points, lines, and areas, by identifying all areas that are within a certain specified distance of a specified object [29]. A buffer analysis with different radii that range from 250 m to 2000 m was used for the first three datasets to determine the total area of each land use class, the length of each road class, and the number of landmarks over the buffering region centered in air quality-monitoring (AQM) stations. Meanwhile, the map of the industrial park was obtained from the Industrial Development Bureau, Taiwan. The map was produced in 2010 and utilized to assess the distance of a point to the nearest industrial park. The DTM with 40-m spatial resolution was provided by the Aerial Survey Office, Forestry Bureau. The DTM represents the topographic altitudes of the study area [30]. Figure 2a shows that the potential predictors from the geographic dataset are the area of each land use class, the length of each road class, the number of landmarks, the distance to the nearest industrial park, and the altitude of a point obtained in different buffer sizes.

*2.3. Satellite Image Data*

Landsat-8 is a satellite launched by the National Aeronautics and Space Administration (NASA) in collaboration with the United States Geological Survey (USGS) that continues its former satellites in the Landsat Data Continuity Mission. The Landsat-8 features an OLI sensor, which provides eight spectral bands with the wavelengths that range from 0.43–1.38 μm, and a thermal infrared sensor that contains two thermal bands with the wavelengths that range from 10.60–12.51 μm. NASA and USGS provide the satellite images in two different product levels, that is, level 1 and level 2. The level 1 product contains digital number information which can be calibrated into radiance or reflectance at the top of the atmosphere, that is, no atmospheric correction is applied. The level 2 product includes radiance or reflectance at the bottom of the atmosphere, or at the Earth's surface, where a built-in atmospheric correction has been implemented. Therefore, the surface radiance or reflectance can represent the true ground information. In this study, the use of the surface reflectance information of Landsat-8 satellite images with a level 2 product is proposed for classification and adopted for the normalized differentiate vegetation index (NDVI) extraction. The classification aims to determine and distinguish the general land uses in satellite images, including vegetation, water bodies, and artificial buildings, while

the NDVI extraction intends to estimate the density and vitality of vegetation. All available Landsat-8 level 2 images acquired from 2013–2018 were used to generate yearly cloud-free images. The cloud covers, which will reduce the classification accuracy, were filtered out from the images in this process. The cloud-free images containing the visible bands and near infrared bands are then utilized to generate the yearly NDVI image by using the formula (NIR-Red)/(NIR+Red), where NIR and Red represent the near infrared and red bands, respectively.

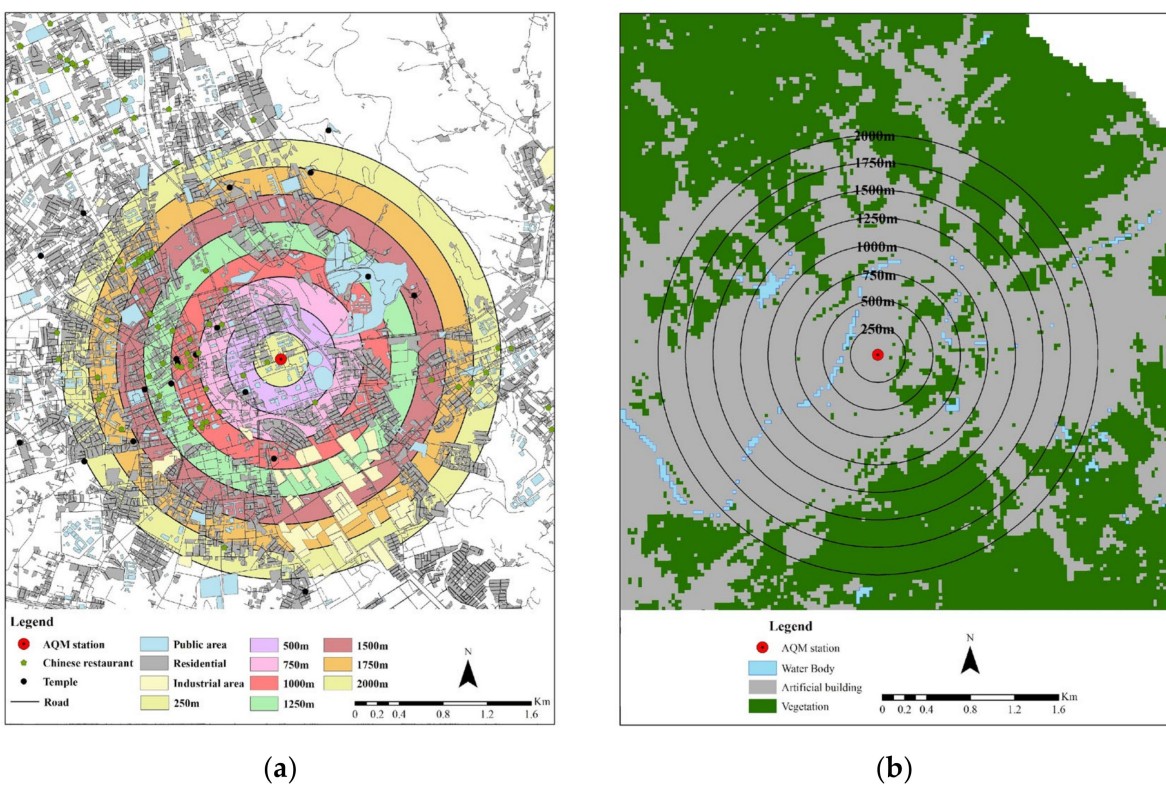

(**a**)                                                                                                        (**b**)

**Figure 2.** Buffering analyses with a radius that ranges from 250 m to 2000 m within Taipei Metropolis. (**a**) Buffering analyses using the potential predictors from a geographic dataset. (**b**) Buffering analyses using satellite image dataset.

The details of the classification and the cloud-free image generation are described in Section 3. Furthermore, the image classification results will be used as predictors in $PM_{2.5}$ modeling. Buffer analysis with a radius of 250 m to 2000 m is used to determine the total area of classification results in each class, that is, the total area of artificial buildings, vegetation, and water bodies. Figure 2b depicts the buffering analysis using the potential predictors $\{B_{250m}, \cdots, B_{2000m}\}$, where $B_{250m}$ represents the first buffering region (radius = 250 m) and $B_{2000m}$ denotes the last buffering region (radius = 2000 m). The predictors of vegetation are denoted as $\{V_{250}, \cdots, V_{2000}\}$, where $V_{250m}$ represents the first buffering region (radius = 250 m), and $V_{2000m}$ denotes the last buffering region (radius = 2000 m). The predictors of water body are denoted as $\{W_{250}, \cdots, W_{2000}\}$, where $W_{250m}$ represents the first buffering region (radius = 250 m), and $W_{2000m}$ denotes the last buffering region (radius = 2000 m).

## 3. Methodology

In this study, a land use classification is applied to a cloud-free image derived from multi-temporal images within a year. Subsequently, the land use image is used in integration with land use classes from a geographic dataset for the estimation of the $PM_{2.5}$

concentration in Taipei metropolis. Section 3.1 describes the generation of cloud-free images for classification, and Section 3.2 defines the $PM_{2.5}$ concentration model's development.

### 3.1. Generation of Cloud-Free Image for Classification

Cloud cover in satellite images is inevitable and hinders the sensor's collection of spectral responses from the Earth surface. Surface reflectance (SR) improves image comparison by accounting for atmospheric effects such as aerosol scattering and thin clouds, which can aid in the detection and characterization of Earth surface changes. The amount of light reflected by the Earth's surface is referred to as surface reflectance. It refers to a surface radiance to surface irradiance ratio that is unitless and ranges from 0–1, and its value increases due to the cloud cover. In addition, if the surface reflectance value and cloud cover are nearly constant and located at different places between two consecutive images, the atmospheric conditions are stable and the cloud cover is not changing significantly over time. This phenomenon could be due to factors, such as the time of day, season, or location [31]. However, it is important to note that cloud cover can vary rapidly and unpredictably over short periods of time, and two images were possibly acquired during different cloud conditions. In addition, the presence of thin clouds or aerosol can affect the accuracy of surface reflectance values because these particles scatter and absorb light differently than clouds, and this can result in errors in the estimation of surface reflectance. To account for these effects and improve the accuracy of surface reflectance values, the use of atmospheric correction algorithms which attempt to remove the effects of atmospheric scattering and absorption from the satellite data, is commonly practiced. These algorithms typically rely on radiative transfer models and atmospheric measurements to estimate the atmospheric conditions and correct the satellite data accordingly. Overall, the accuracy of surface reflectance values can be affected by various factors, including cloud cover, atmospheric conditions, and sensor calibration. The careful evaluation of these factors is important when comparing satellite images and interpreting changes in the Earth's surface over time.

To deal with the cloud covers in satellite images and to generate a cloud-free image, a simple cloud detection with a pixel replacement strategy is performed. The basic idea is to composite a cloud-free image from a set of cloudy images over a year. Specifically, the reflectance values of cloud pixels in a satellite image are replaced by the corresponding cloud-free pixels in other images (i.e., the same position but a different acquisition time). For simplicity, the pixel with the lowest reflectance value is regarded as a cloud-free pixel (i.e., a low probability of being a cloud pixel) and is used to fill in that position. This process is applied to each pixel in the image, and a nearly cloud-free image is generated. Notably, this process did not guarantee that the generated image would be cloud-free, and a near cloud-free image is sufficient for land use classification.

After the image composition process, the maximum likelihood classification is adopted for land use classification. The classification employs the statistical features of the data. The mean and standard deviation of each spectral band and textural indices of the image are computed first, and the likelihood of each pixel that belongs to individual classes is then computed using some classical statistics and probabilistic relationships and a normal distribution for the pixels in each class. Finally, the pixels are assigned to a class of features based on their likelihood. In this study, the training and testing datasets use the surface reflectance of all spectral bands as the input and its land use class as the label, which is manually selected by visual inspection. The workflow of the cloud-free image generation and land use classification is shown in Figure 3.

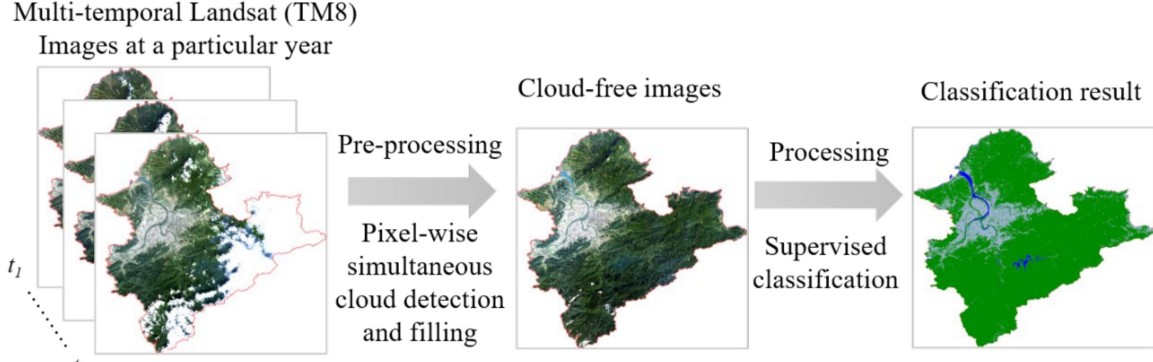

**Figure 3.** Workflow of cloud-free image generation and classification. This process involves multiple steps, including image acquisition, pre-processing and analysis, *n* represents the number of used images.

*3.2. PM$_{2.5}$ Concentration Model Development*

The modeling of PM$_{2.5}$ concentration consists of two successive steps: a Spearman correlation (r) analysis and stepwise variable selection, as illustrated in Figure 4. The correlation analysis is performed to examine the bivariate relationship between PM$_{2.5}$ concentrations and potential predictors, including the air quality parameters from the air quality dataset and the predictors from the geographic and satellite image datasets discussed in Section 2.2. The analysis concerns the magnitude and direction of correlation between the PM$_{2.5}$ concentration and a potential predictor. The correlation direction means the positive or negative relationship between the two variables. A predictor is eliminated when its direction is incorrect. The correctness of the direction of each predictor is based on a priori knowledge of its analytic relationship to the PM$_{2.5}$ concentration. For instance, suppose that the two variables are PM$_{2.5}$ concentration and vegetation, where the vegetation is likely to absorb the PM$_{2.5}$ exposure. Then, the existence of vegetation can reduce the concentration of PM$_{2.5}$, that is, they should have negative correlation. If the determination of correlation based on the dataset is opposite, then the vegetation is further removed. In addition, the magnitude of correlation represents the strength of positive or negative relationships between variables. Therefore, the thresholds to determine the suitable strength relationship between the PM$_{2.5}$ concentration and each potential predictor are r $\leq -0.4$ and r $\geq +0.4$, respectively. The potential predictors that did not satisfy one of the two concerns will not enter the next part.

In the stepwise variable selection, the potential predictors from the previous part were included in the model development and further evaluated based on the *p*-value and variance inflation factor (VIF). The *p*-value and VIF indicate the significance of a particular predictor to the model and the multi-collinearity between that and the other predictors, respectively. The *p*-value ranges from 0 to 1, whereas VIF ranges from 0 to infinite. A value close to 0 in both measures means that the particular predictor is more significant and has less multi-collinearity. Therefore, predictors with a *p*-value smaller than 0.1 are kept and reselected for the model development, whereas the opposite is removed from the model. The reselection and elimination based on the *p*-value are repeated until the potential predictors in the model satisfy the *p*-value threshold. Furthermore, the VIF of potential predictors is determined, and those predictors with VIF larger than 3 were kept and set as the final predictors for the estimation of PM$_{2.5}$ concentration. The formulation of the estimation is expressed as follows:

$$PM_{2.5} = \beta_0 + \beta_1 \times X_1 + \ldots + \beta_n \times X_n, \tag{1}$$

where PM$_{2.5}$ is the concentration of PM$_{2.5}$ exposure; $\beta_0$ represents the regression bias; and $[X_1, \ldots, X_n]$ and $[\beta_1, \ldots, \beta_n]$ are the n final predictors and their corresponding regression

coefficients, respectively. To develop the model, 90% of air quality-monitoring sites were utilized, while the remaining 10% were employed for evaluation. In order to evaluate the model, the root mean square error (RMSE), and ordinary and adjusted coefficients of determination ($R^2$) were used. The evaluation process was carried out using a 10-fold cross-validation approach to determine the accuracy and performance of the model. Additionally, a second methodology was utilized to verify the accuracy of the model, where data collected in 2018 were considered as out-of-sample data, and the model's precision was confirmed by estimating $PM_{2.5}$ concentrations from out-of-sample observations and comparing them to known observations.

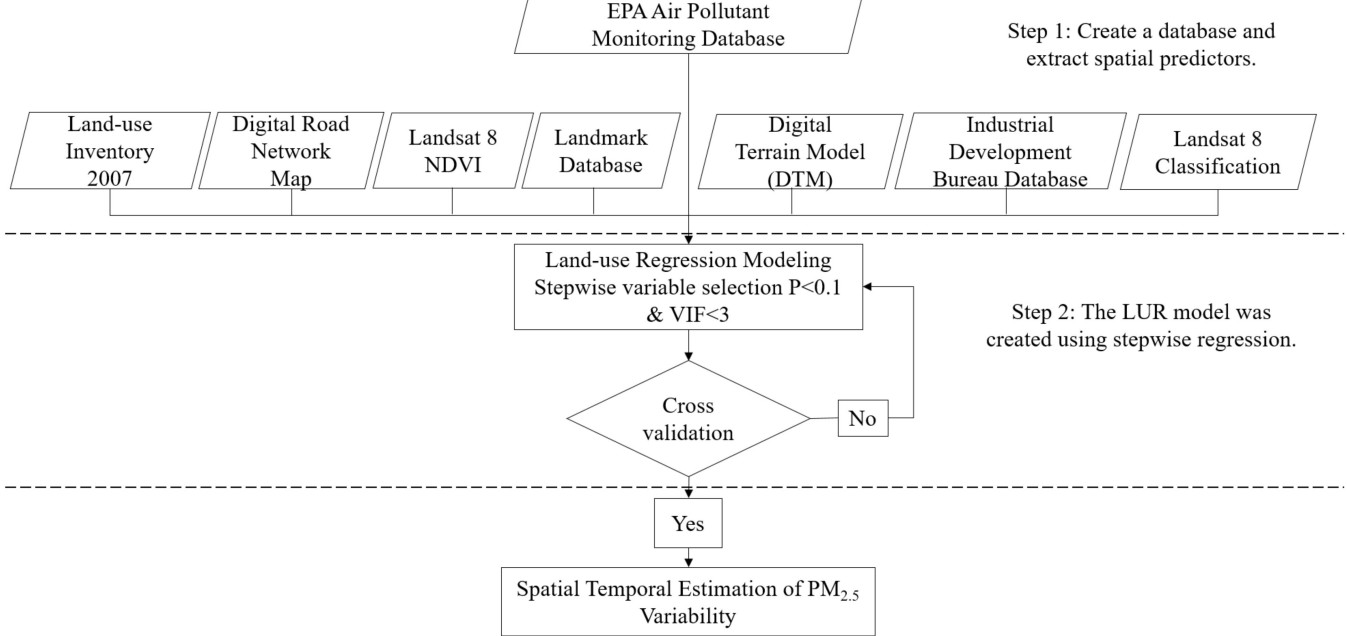

**Figure 4.** Framework of model development and evaluation for $PM_{2.5}$ concentration estimation.

## 4. Experimental Result and Discussion

This study proposed to utilize and integrate the land use information in geographic and remote sensing datasets along with air quality parameters, including the concentrations of $O_3$, $PM_{10}$, CO, $SO_2$, and $NO_2$ collected from 17 stations from 2013–2018, to model the $PM_{2.5}$ concentration in the Taipei Metropolis. The land cover information in the remote sensing dataset was obtained by applying a maximum likelihood classification to the generated cloud-free images. The results of cloud-free image generation and the image classification are presented in Section 4.1. The estimation of $PM_{2.5}$ concentrations using the proposed method is described in Section 4.2. The performance of the developed model was compared with the related methods, that is, the models using land use information in the geographic and remote sensing datasets, separately. The comparison results are reported and discussed in Section 4.3.

### 4.1. Cloud-Free Image and Classification

One of the main challenges for satellite image applications is the existence of cloud cover that hinders partial ground information. To address this issue, several cloudy images are merged to generate a near cloud-free image. Figure 5 shows the original satellite images and cloud-free images that were acquired from 2013–2018, in which cloud cover was spotted at several places. High coverage of clouds can be found in the images of 2013, 2014, and 2016. Therefore, three satellite images were used in each of these years to obtain cloud-free images with satisfactory results, and only two images were utilized for the generation of cloud-free images in the other years. With a simple merging strategy, the cloud cover is significantly reduced, and near cloud-free images are generated.

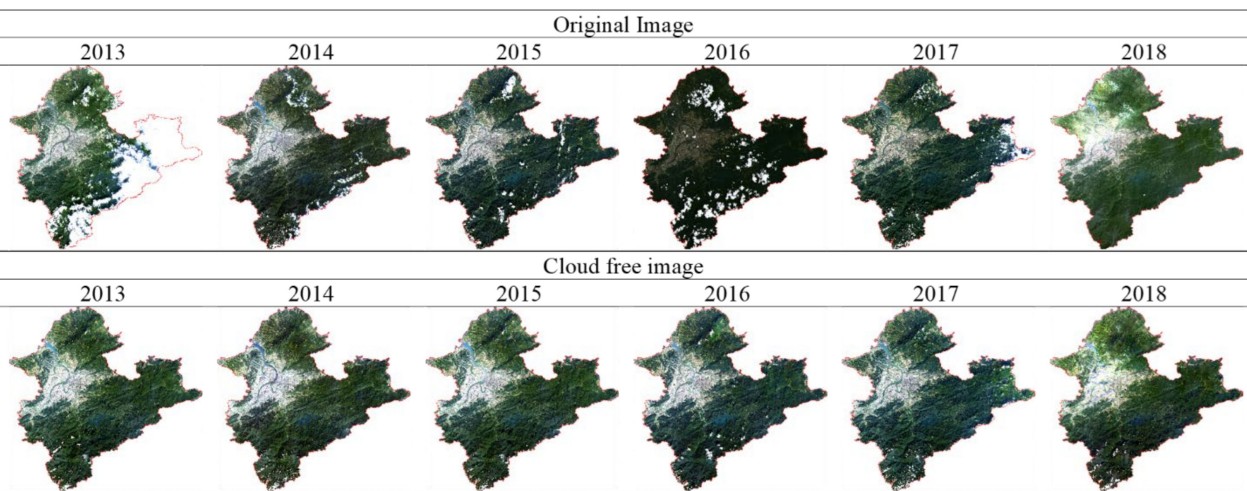

**Figure 5.** Original (**top**) and cloud-free satellite images (**bottom**) from 2014 to 2018.

The near cloud-free images were used to extract land use information in Taipei Metropolis by using maximum likelihood classification. The selected land use types are water bodies, vegetation, and artificial building. The training and testing sets for the classification were determined by using manual visual selection. Figure 6 shows the classification results, and Figure 7 presents the trend of land use coverage changes in terms of total coverage area in the unit of Ha during 2013 to 2018. The results show that the coverage of artificial buildings is gradually decreasing, whereas that of vegetation is gradually increasing, except for the year 2015. This phenomenon may be caused by several parks that were created during those six years. As for the cover of water bodies, the total coverage area slightly decreased.

To evaluate the classification results, a confusion matrix is calculated and shown in Table 2. Notably, the indicators of accuracy in matrix confusion are overall accuracy and the Kappa coefficient. These two indicators represent the ratio between the numbers of correct classified pixels to the numbers of training and testing pixels, respectively. From the statistical numbers in Table 1, the overall accuracy is more than 95%, and the Kappa coefficient is higher than 90%, indicating that the classification results are suitable for the generation of $PM_{2.5}$ concentration predictors.

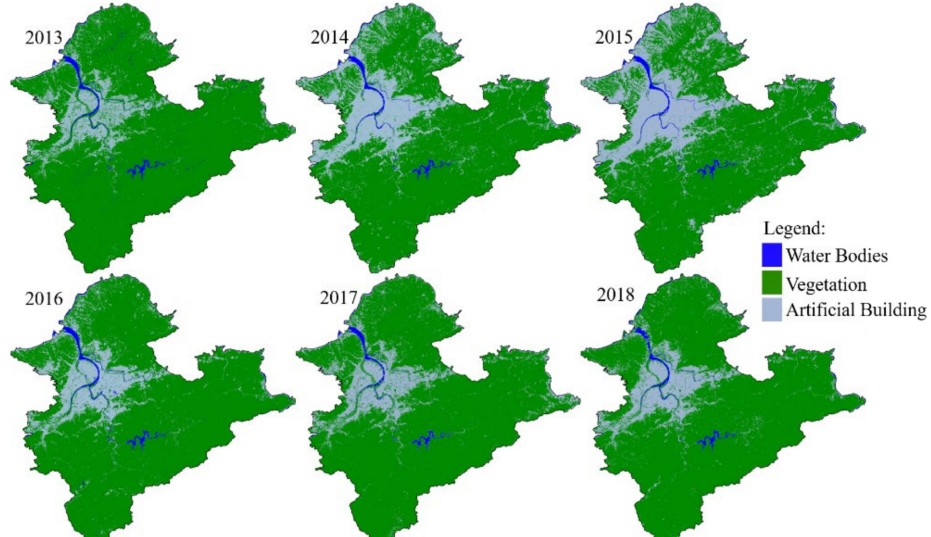

**Figure 6.** Land coverage information from satellite images using maximum likelihood classification.

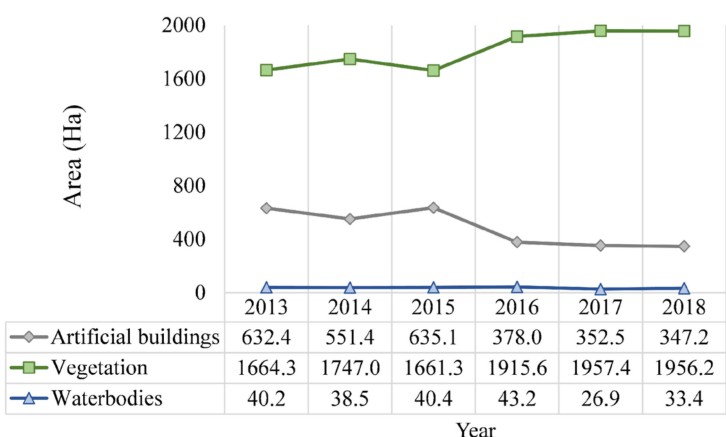

**Figure 7.** Land cover distribution area and their changes during 2013–2018.

**Table 2.** Accuracy of satellite image land cover classification. P.A. and U.A. represent the producer and user accuracies, respectively.

| Land Use Type | 2013 | | 2014 | | 2015 | | 2016 | | 2017 | | 2018 | |
|---|---|---|---|---|---|---|---|---|---|---|---|---|
| | P.A. | U.A. | P.A. | U.A. | P.A. | U.A. | P.A. | U.A. | P.A. | U.A. | P.A. | U.A. |
| | in Percentage (%) | | | | | | | | | | | |
| Artificial buildings | 99.9 | 89.3 | 99.6 | 94.9 | 99.9 | 91.7 | 96.8 | 97.5 | 93.2 | 88.9 | 97.4 | 84.1 |
| Vegetation | 96.3 | 99.9 | 98.5 | 99.9 | 97.2 | 99.9 | 99.4 | 98.8 | 96.7 | 98.0 | 94.8 | 99.2 |
| Water bodies | 98.7 | 100.0 | 98.2 | 100.0 | 98.7 | 100.0 | 98.0 | 100.0 | 100.0 | 100.0 | 100.0 | 100.0 |
| Overall acc. | 97.4 | | 98.70 | | 98 | | 98.70 | | 96 | | 95.5 | |
| Kappa coef. | 95 | | 98 | | 96 | | 98 | | 90 | | 90 | |

*4.2. PM$_{2.5}$ Concentration Model with Geographic and Satellite Image Integration*

The development of the estimation model for PM$_{2.5}$ concentration in Taipei Metropolis starts from analyzing the correlation between PM$_{2.5}$ exposure and the potential predictors by means of Spearman correlation. Table 3 provides the Spearman correlation between the PM$_{2.5}$ exposure and the land cover predictors from the satellite image datasets. The statistic numbers reveal that the predictors of artificial buildings $\{B_{250m}, \cdots, B_{2000m}\}$ and vegetation $\{V_{250m}, \cdots, V_{2000m}\}$ show strong positive and negative correlations, respectively. Therefore, these two sets of predictors are included for further selection by using stepwise variable selection. Meanwhile, for those from the class of waterbodies, only three predictors with weak correlation, namely, $W_{250m}$, $W_{500m}$, and $W_{750m}$, were selected.

A similar process is applied to the potential predictors from the geographic dataset. A total of 50 predictors, namely, PM$_{10}$, NO$_x$, NO$_2$, wind speed, ambient temperature, relative humidity, NDVI$\{ndvi_{250m}, \cdots, ndvi_{2000m}\}$, pure residential $\{pR_{500m}, pR_{750m}\}$, mixed residential $\{mR_{500}\}$, expressway $\{EW_{1250m}, EW_{1500m}, EW_{1750m}, EW_{2000m}\}$, major road $\{MR_{500,}\}$, local road $\{LR_{500}\}$, industrial-commercial residential $\{iR_{500,} iR_{750,}\}$, elevation $\{DTM\}$, airport within distance, artificial building predictors $\{B_{250m}, \cdots, B_{2000m}\}$, vegetation predictors $\{V_{250m}, \cdots, V_{2000m}\}$, and water body predictors $\{W_{250m}, W_{500m}, W_{750m}\}$, satisfy the correlation criterion. These 50 predictors are further utilized and evaluated in the stepwise variable selection. Table 4 shows the estimation model for PM$_{2.5}$ concentrations with five final predictors, which are selected after the evaluation in the stepwise variable selection. Three of the final predictors, namely, the concentration of PM$_{10}$, wind speed, and the concentration of SO$_2$, are obtained from the air quality dataset. The remaining predictors include $B_{2000m}$ (the predictor of artificial building with the radius = 2000 m) and $pR_{500m}$ (the predictor of pure residential with the radius = 500 m), which are obtained from the satellite image and geographic datasets, respectively. In the buffering analysis, these two

predictors represent the total area of artificial buildings and pure residential areas with radius = 2000 m and 500 m, respectively.

**Table 3.** Spearman correlation between $PM_{2.5}$ exposure and the potential predictors from the satellite image datasets. The buffering analysis is denoted as BA.

| Class | Potential Predictors | | r |
|---|---|---|---|
| | Radius in BA | Notation | |
| Artificial buildings | 250 m | $B_{250m}$ | 0.57 |
| | 500 m | $B_{500m}$ | 0.63 |
| | 750 m | $B_{750m}$ | 0.65 |
| | 1000 m | $B_{1000m}$ | 0.67 |
| | 1250 m | $B_{1250m}$ | 0.66 |
| | 1500 m | $B_{1500m}$ | 0.65 |
| | 1750 m | $B_{1750m}$ | 0.66 |
| | 2000 m | $B_{2000m}$ | 0.66 |
| Water bodies | 250 m | $W_{250m}$ | −0.19 |
| | 500 m | $W_{500m}$ | −0.02 |
| | 750 m | $W_{750m}$ | −0.04 |
| | 1000 m | $W_{1000m}$ | 0.06 |
| | 1250 m | $W_{1250m}$ | 0.16 |
| | 1500 m | $W_{1500m}$ | 0.05 |
| | 1750 m | $W_{1750m}$ | −0.19 |
| | 2000 m | $W_{2000m}$ | −0.02 |
| Vegetation | 250 m | $V_{250m}$ | −0.56 |
| | 500 m | $V_{500m}$ | −0.63 |
| | 750 m | $V_{750m}$ | −0.66 |
| | 1000 m | $V_{1000m}$ | −0.69 |
| | 1250 m | $V_{1250m}$ | −0.55 |
| | 1500 m | $V_{1500m}$ | −0.70 |
| | 1750 m | $V_{1750m}$ | −0.71 |
| | 2000 m | $V_{2000m}$ | −0.71 |

**Table 4.** Coefficient of predictors in the final model for $PM_{2.5}$ concentrations estimation.

| Predictor | $\beta$ | *p*-Value | VIF | Part. $R^2$ | $R^2$ | Adj. $R^2$ | RMSE |
|---|---|---|---|---|---|---|---|
| $\beta_0$ (Intercept) | 2.03 | 0.26 | - | - | 0.85 | 0.72 | 3.06 |
| $PM_{10}$ | 0.28 | <0.001 | 1.67 | 70.7% | | | |
| $B_{2000m}$ | $0.01 \times 10^{-3}$ | <0.001 | 1.41 | 21.2% | | | |
| Wind speed | −1.00 | <0.01 | 1.01 | 12.2% | | | |
| $SO_2$ | 1.86 | <0.01 | 1.82 | 7.4% | | | |
| $pR_{500m}$ | $-0.71 \times 10^{-4}$ | <0.1 | 1.01 | 0.7% | | | |

The adjusted $R^2$ = 0.72 implies that the final model has a high positive correlation with the $PM_{2.5}$ concentrations. The regression coefficients of the predictors $B_{2000m}$ and $pR_{500m}$ are $0.01 \times 10^{-3}$ and $-0.71 \times 10^{-4}$, indicating that the $PM_{2.5}$ concentration is increased by $0.01 \times 10^{-3}$ µg/m$^3$ when the area inside the circular buffer with a radius of 2000 m is all artificial building, and is decreased by $0.71 \times 10^{-4}$ µg/m$^3$ if the land use class is pure residential in a circular buffer with a radius of 500 m. Moreover, the $PM_{10}$ concentration can dominantly explain the variation in $PM_{2.5}$ concentration because the partial (part.) $R^2$ is 70.7%. For the other selected predictors, such as $B_{2000m}$, wind speed, the $SO_2$ concentration, and $pR_{500m}$, the part. $R^2$ is 21.2%, 12.2%, 7.4%, and 0.7%, respectively. In addition, the $R^2$ of the developed model is 0.85, and the scatterplot of the observed and estimated $PM_{2.5}$ concentrations is shown in Figure 8.

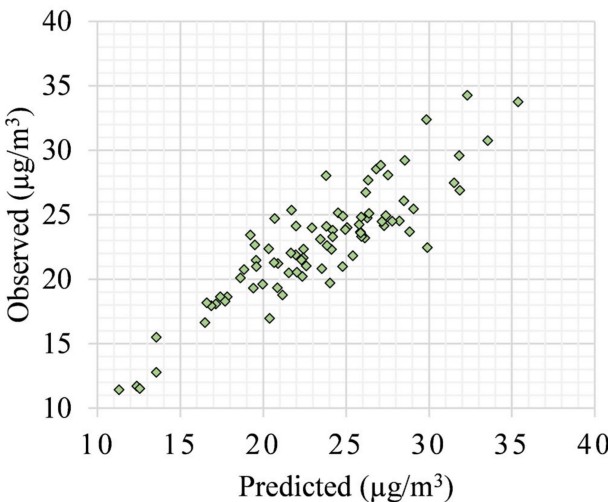

**Figure 8.** Scatterplot of observed (*y*-axis) and estimated (*x*-axis) PM$_{2.5}$ concentration. The PM$_{2.5}$ estimation model developed by using land use information from the geographic and satellite image datasets.

Furthermore, a tenfold cross-validation was performed to evaluate the developed model for the estimation of PM$_{2.5}$ concentrations. Approximately 10 to 11 samples at each fold were used to determine model accuracy. The results are shown in Figure 9. On average, the RMSE at all folds is 3.06 µg/m$^3$, whereas the R$^2$ is 0.76. They are in line with the accuracy obtained from the model, that is, the RMSE and R$^2$ are 3.06 µg/m$^3$ and 0.85, respectively. This finding indicates that the developed model is effective in estimating PM$_{2.5}$ concentrations. Subsequently, the developed model is used to estimate the PM$_{2.5}$ concentrations of the entire Taipei Metropolis from 2013 to 2018. The results are shown in Figure 10. The colors that range from yellow to red represent the lowest (5 µg/m$^3$) and highest (30 µg/m$^3$) concentrations of PM$_{2.5}$, respectively. Comparing Figures 6 and 10, the red colors are mostly located on the areas in which the land use class is artificial building, and the yellowish colors are located in the areas of vegetation cover. PM$_{2.5}$ pollution levels can be observed to exhibit fluctuating trends over time due to various factors such as weather pattern, human activities, and natural events. For instance, in regions with high traffic, industrial activity or during specific seasons, PM$_{2.5}$ levels may be consistently higher than in other areas or at other times. Therefore, it is essential to consider these factors and analyze the cause of the observed fluctuations in PM$_{2.5}$ levels. To verify the accuracy of the model used for PM$_{2.5}$ concentration estimation, a comparison of the predicted PM$_{2.5}$ concentration with the actual measurements was performed (Figure 8). By comparing the predicted and actual PM$_{2.5}$ concentrations, the accuracy of the model and its effectiveness in predicting PM$_{2.5}$ levels could be assessed.

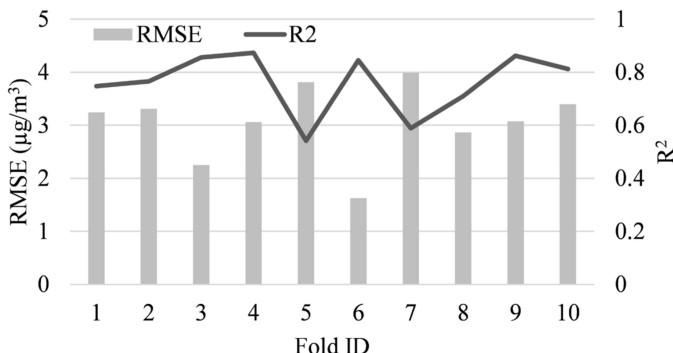

**Figure 9.** Accuracy assessment of the developed model using tenfold cross-validation. The measurement RMSE (**left**) and R$^2$ (**right**) are used.

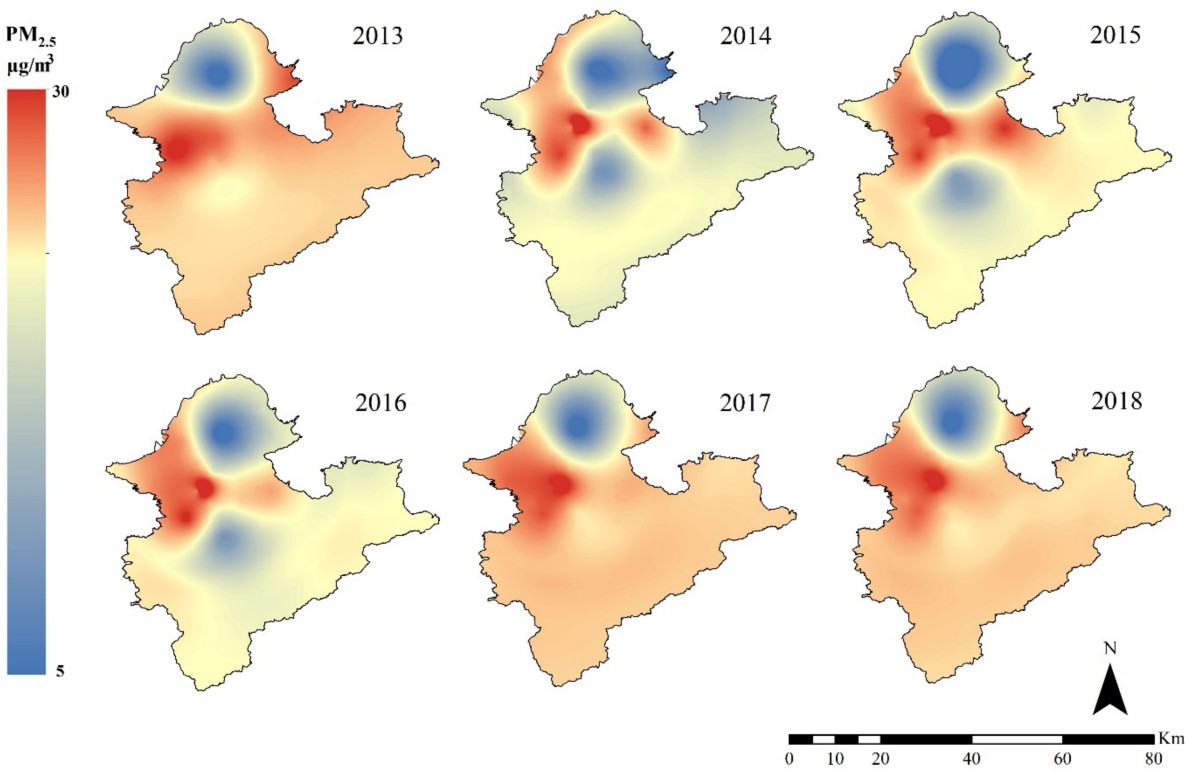

**Figure 10.** Prediction maps of the spatial–temporal variability of $PM_{2.5}$ concentration using the developed model.

### 4.3. Comparison with Related Models

One of the contributions of this study is to integrate the land use information of the geographic and satellite image datasets into one model for the estimation of $PM_{2.5}$ concentration. To evaluate this contribution, the model developed using the land use information from both datasets was compared with the models that use predictors from a single dataset. The data preprocessing, correlation analysis, and stepwise variable selection are the same as those from the three compared models. The difference is that the input to the compared models is the land use predictors from the satellite image dataset (denoted as Model A) or geographic dataset (denoted as Model B), that is, the two datasets are used separately.

In Model A, 32 potential predictors, including the land cover predictors from the satellite image dataset and the air quality predictors from the air quality dataset, are reported. After the correlation analysis by means of Spearman correlation, 26 predictors, namely, $SO_2$, CO, $CO_3$, $PM_{10}$, $NO_x$, NO, $NO_2$, artificial building $\{B_{250m}, \cdots, B_{2000m}\}$, vegetation $\{V_{250m}, \cdots, V_{2000m}\}$, and water body $\{W_{250m}, W_{500m}, W_{750m}\}$, are selected from the elimination. Furthermore, four predictors, $PM_{10}$ concentration, $SO_2$ concentration, $B_{2000m}$, and $W_{250m}$, satisfy the thresholds in the stepwise variable selection. Meanwhile, in Model B, 68 predictors consisting of the land use predictors from the geographic dataset and the air quality predictors from air quality dataset were accepted after the correlation analysis, and they are further considered in the stepwise variable selection. After the stepwise variable selection, four predictors, $PM_{10}$ concentration, $SO_2$ concentration, wind speed, and $pR_{1500m}$, are selected, and they are set as the input to the estimation model.

Tables 5 and 6 present the statistical details of the comparison of Models A and B, respectively. As expected, the $PM_{10}$ concentration played a significant role in the estimation of $PM_{2.5}$ concentrations (part. $R^2 = 59.0\%$ in Model A, part. $R^2 = 33.2\%$ in Model B). This finding is in line with the developed model, in which the part. $R^2$ is 70.7%. For accuracy, the $R^2$ and RMSE in Model A are 0.68 and 3.14 μg/m³, and those for Model B are 0.79 and 3.47 μg/m³, respectively. The $R^2$/RMSE in these two compared models are lower/higher

than those of the developed model. This result indicates that the integration of land use predictors in the geographic and satellite image datasets can increase the accuracy of $PM_{2.5}$ concentration modeling. In addition, Figures 11 and 12 show the comparison between the observed and estimated $PM_{2.5}$ concentrations along with their results in the tenfold cross-validation from the two compared models, respectively. Furthermore, the precision of the proposed model is compared with two alternative models, as model A and model B, by analyzing the regressed prediction data from the year 2018, as depicted in Figure 13. The results presented in Figure 13a demonstrate the effectiveness of the proposed model by showcasing its superior performance in comparison to Model A and Model B. This has been established through a linear regression analysis, which has revealed a notably higher R-squared value for the proposed model. These results serve as a strong indication of the efficacy of the proposed algorithm, and its potential to enhance the predictive capabilities of existing models. The findings presented here may have implications for future research, particularly in the field of predictive modeling, wherein the development of more effective models is of critical importance.

**Table 5.** Predictor coefficients in Model A for $PM_{2.5}$ concentration estimation.

| Predictor | $\beta$ | $p$-Value | VIF | Part. $R^2$ | $R^2$ | Adj. $R^2$ | RMSE |
|---|---|---|---|---|---|---|---|
| $\beta_0$ (Intercept) | 1.29 | 0.43 | - | - | 0.68 | 0.7 | 3.14 |
| $PM_{10}$ | 0.29 | <0.001 | 1.80 | 59.0% | | | |
| $B_{2000m}$ | $0.01 \times 10^{-3}$ | <0.001 | 1.56 | 13.0% | | | |
| $SO_2$ | 1.75 | <0.01 | 1.82 | 8.0% | | | |
| $W_{250m}$ | $-0.01 \times 10^{-2}$ | <0.01 | 1.18 | 0.7% | | | |

**Table 6.** Predictor coefficients in Model B for $PM_{2.5}$ concentration estimation.

| Predictor | $\beta$ | $p$-Value | VIF | Part. $R^2$ | $R^2$ | Adj. $R^2$ | RMSE |
|---|---|---|---|---|---|---|---|
| $\beta_0$ (Intercept) | 0.55 | 0.8 | - | - | 0.79 | 0.62 | 3.47 |
| $PM_{10}$ | 0.33 | 0.001 | 1.62 | 33.2% | | | |
| $SO_2$ | 3.25 | 0.001 | 1.61 | 16.9% | | | |
| Wind Speed | $-1.12$ | 0.002 | 1.02 | 7.4% | | | |
| $pR_{1500m}$ | $0.01 \times 10^{-1}$ | 0.01 | 1.04 | 5% | | | |

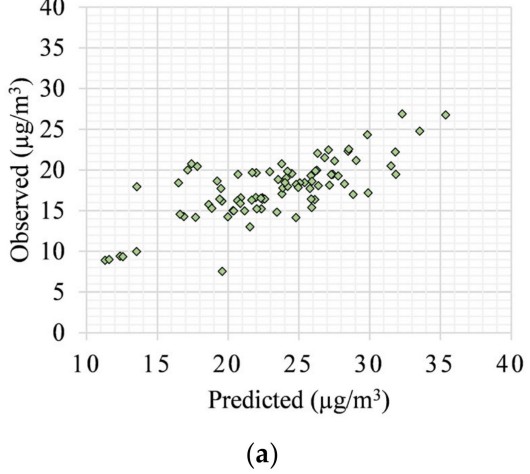 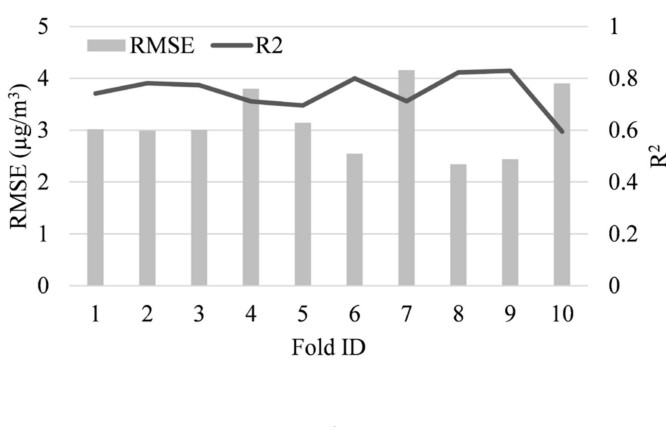

(**a**) (**b**)

**Figure 11.** External verification using the $PM_{2.5}$ data of the year 2018. (**a**) Observed (*y*-axis) and estimated (*x*-axis) $PM_{2.5}$ concentration from Model A. (**b**) Accuracy assessment using tenfold cross-validation.

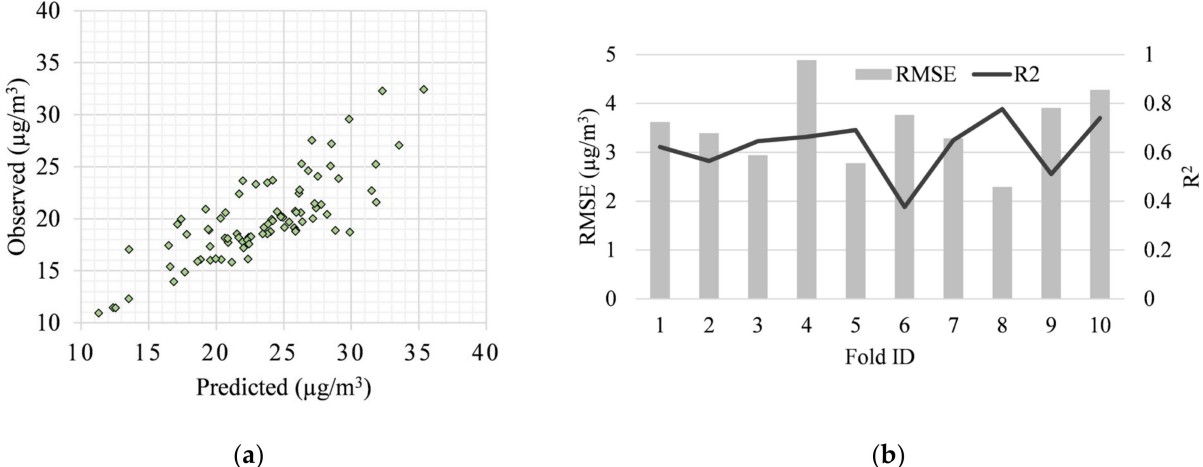

**Figure 12.** External verification using the PM$_{2.5}$ data of the year 2018. (**a**) Observed (*y*-axis) and estimated (*x*-axis) PM$_{2.5}$ concentration from Model B. (**b**) Accuracy assessment using tenfold cross-validation scheme.

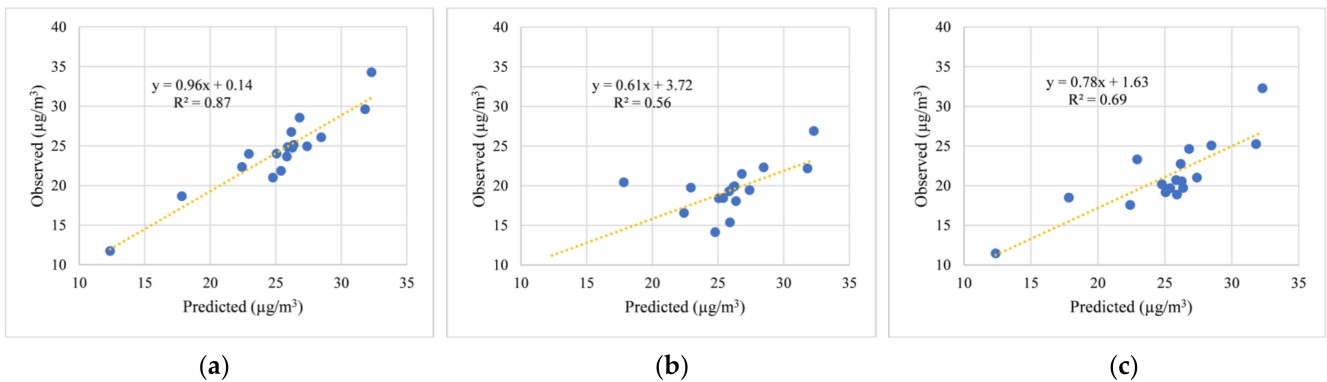

**Figure 13.** The relationship between the observation and the predictors is evaluated through regression analysis in 2018. (**a**) Observation regressed against predictors with integration between geographic and remote sensing datasets. (**b**) Observation regressed against predictors with a remote sensing dataset. (**c**) Observation regressed against predictors with a geographic dataset.

## 5. Conclusions

The contributions of this study include the integration of land use classes from geographic and satellite image datasets into an estimation model for PM$_{2.5}$ concentration, the use of Landsat-8 images with better spatial resolution, and the generation of cloud-free images for land use classification using satellite images. The model was developed by means of predictor selection, which was based on two successive analyses, correlation analysis and stepwise variable selection. Among all the potential predictors, only five of them, including the PM$_{10}$ concentration, wind speed, and the SO$_2$ concentration from air quality datasets, as well as $B_{2000m}$ and $pR_{500m}$ from the satellite image and geographic dataset, survived the selection. The last two predictors indicate the total area of artificial buildings and pure residential areas under a circular buffer of 2000 m and 500 m, and represent the integration of geographic and satellite image datasets. Based on their accuracy assessment, the RMSE and R$^2$, which were 3.06 µg/m$^3$ and 0.85, respectively, imply that the developed model can estimate the PM$_{2.5}$ concentration with satisfactory accuracy. Moreover, the developed model was compared with two related models that utilized land use classes, either from the geographic dataset or from the satellite image dataset separately. The comparison shows that the developed model outperformed the two related models, that is, the integration between land use classes from the geographic and satellite image datasets

can improve the performance of the estimation model. Using the model that was created by integrating geographic and satellite image datasets, the PM$_{2.5}$ levels of Taipei Metropolis between 2013 and 2018 were estimated. The concentration of PM$_{2.5}$ was depicted using colors ranging from yellow to red, with the former indicating the lowest (5 $\mu$g/m$^3$) and the latter the highest (30 $\mu$g/m$^3$) concentrations. In the future, the potential predictors from the satellite image dataset can be extended in more detail (i.e., by adding other classes such as agricultural, bare land, shrub, and bush land). Therefore, the estimation of PM$_{2.5}$ concentrations may improve.

**Author Contributions:** Conceptualization, D.H., C.-D.W. and C.-H.L.; Methodology, D.H. and C.-D.W.; Software, D.H.; Validation, D.H. and M.A.S.; Formal analysis, D.H. and M.A.S.; Investigation, D.H.; Resources, D.H. and C.-D.W.; Data curation, D.H. and C.-D.W.; Writing—original draft preparation, D.H. and M.A.S.; Writing—review and editing, D.H. and C.-H.L.; Visualization, D.H.; Supervision, C.-H.L.; Project administration, C.-H.L.; Funding acquisition, C.-H.L. All authors have read and agreed to the published version of the manuscript.

**Funding:** This research and the APC were funded by Ministry of Science and Technology, Taiwan (grant numbers MOST 111-2121-M-006-012).

**Data Availability Statement:** Not applicable.

**Conflicts of Interest:** The authors declare no conflict interest.

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
