# Peer review of "Analysis of Spatial–Temporal Variability of PM2.5 Concentrations Using Optical Satellite Images and Geographic Information System"

_remotesensing, doi:10.3390/rs15082009_

Round 1

Reviewer 1 Report

General comments:

1. The authors must specify whether the validation was conducted using independent data or was incorporated into the model.

2. In order to assess the efficacy of the proposed model for PM2.5 concentration, it is necessary to conduct a thorough analysis comparing the accuracy of the proposed method to that of other approaches that make use of comparable data and methodologies.

a. Correct the sentence (page-2, line 86)

The remote-sensing sensor is altered from MODIS to Landsat-8 operational land imager (OLI) sensor.

b. Correct the spelling error (page-4, line 189):

The predictors of water body are dented as ...

c. Check the sentence (page 6, line 205)

In addition, the surface reflectance value and the cloud covers are nearly constant and located at different places in between two images which were acquired at consecutive acquisition times.

1. What is the main question addressed by the research?

Using point data and other ground and satellite datasets, the article proposes a model to estimate the PM2.5 concentration across a region.

2. Do you consider the topic original or relevant in the field? Does it address a specific gap in the field?

There are few stations that measure air quality parameters such as PM2.5. Consequently, using ground and satellite data, the proposed model can be used to generate a PM2.5 distribution map.

3. What does it add to the subject area compared with other published material?

According to the authors, the performance of the proposed procedure in terms of accuracy is superior to other approaches. Authors may require additional references to substantiate their claim.

4. What specific improvements should the authors consider regarding the methodology? What further controls should be considered?

Authors may compare their findings to other methodologies. In the conclusion, the interval at which such PM2.5 maps can be constructed may be suggested.

5. Are the conclusions consistent with the evidence and arguments presented and do they address the main question posed?

Authors may need to critically compare their results with other approaches.

6. Are the references appropriate?

More references are needed to compare the results of this study with others.

7. Please include any additional comments on the tables and figures.

In the conclusion section, authors may present the list of predictors that were found to be beneficial for modeling PM2.5 over unknown locations/regions.

Author Response

We would like to give our sincere thanks to Editor and all ‘reviewers’ detailed and insightful comments as well as suggestions to improve the paper. We appreciate all the remarks regarding the presentation and approach description and are happy to incorporate them into this revised manuscript. In addition, the manuscript was proofread by a native speaker. Please refer to the accompanying documents.

Your kind decision on this paper will be very much appreciated.

Sincerely,

Chao-Hung Lin

Reviewer 2 Report

The authors used geospatial datasets and multispectral satellite data, and used Spearman correlation analysis and stepwise variable selection to screen factors and establish spatial regression relationships to analyze PM2.5 concentrations. Additionally, a cloud-free image was constructed for land classification. This article has some innovative aspects, but there are also some issues:

1. The numbering of figures and tables in the manuscript is incorrect. Please correct them.

2. The organization and expression of the manuscript read more like a thesis than an article. It would be helpful to focus on more useful information, and not include too many details on well-known or less important content. I hope this can be improved.

3. In figures 1 and 7, the indicators show that "PM2.5 levels are decreasing year by year as vegetation increases and artificial building decreases". However, section 4.2 and "figure 7" indicate that PM2.5 pollution has a fluctuating trend. It would be helpful to explain the reasons for this fluctuation while verifying the accuracy of the model.

4. The main experiment and the comparison experiment have different sets of variables selected by the model factor selection mechanism. However, simply comparing the accuracy metrics of the two experiments is not convincing enough. For example, the pR500m factor was selected in the main experiment, but did not appear in the selection for the comparison experiment. According to the logic of the main experiment, the pR500m factor can better explain the cause of PM2.5. It is necessary to explain the scientific validity of the comparison experiment and the reasons for not selecting the pR500m factor or selecting new factors, and provide explanations.

5. The paper emphasizes "buildings, vegetation, and water bodies" for the most part in the early stages, and the vegetation factor shows a strong negative correlation in the article, but there is no vegetation factor in the further experiment, which is confusing. Please explain the reason or make modifications.

6. The analysis in the article is insufficient. The formation and changes of PM2.5 concentration were not analyzed in depth, and only the model accuracy was compared, deviating from the topic.

7. The title of the article only emphasizes the use of satellite images, but in fact, it also incorporates information from geographical data sets, which is not well reflected in the title. I suggest modifying the title.

8. The article did not demonstrate the advantages of the proposed method over commonly used PM2.5 prediction methods. It is recommended to add relevant comparative experiments.

9. Add visual comparison experiments to the experimental results and analyze the rationality and accuracy of the proposed model.

Author Response

(The authors gave the same response as above.)

Reviewer 3 Report

Thank you for giving me this opportunity to read the manuscript entitled "Analysis of Spatial-temporal Variability of PM2.5 Concentrations using Optical Satellite Images". The topic of this manuscript is interesting and would be a good contribution to this field. I think it could be considered for publication in RS once the following issues are addressed.

1.     Please replace the keywords that already appear in the manuscript's title with close synonyms or other keywords, which will also facilitate your paper being searched by potential readers.

2.     Root mean square error (RMSE), relative root mean square error (%RMSE), and mean absolute error (MAE) should be used to evaluated the regression model. The coefficient of determination (r2) is not a good index to evaluate the performance of your model.

3.     Spatio-temporal heterogeneity of the PM2.5-landuse relationship should be well considered in the proposed model.

4.     Line 35, “In the worst cases where the exposure of PM2.5 is high [5, 6], …”: a paper titled “Dynamic assessment of PM2. 5 exposure and health risk using remote sensing and geo-spatial big data” is suggested to be added as a reference to support the statement here,

5.     Some grammatical errors exist in the manuscript. Therefore, a critical review of the manuscript's language will improve its readability.

Author Response

(The authors gave the same response as above.)

Round 2

Reviewer 2 Report

The author has made significant efforts in revising the manuscript, highlighting the core content and the validity of the experiment. Most of the previously raised issues have been satisfactorily addressed. However, there are still some remaining issues as follows: The temporal resolution of the experiment is one year, so it is necessary to explain the processing steps of the land surface index (like NDVI) extracted from the optical satellite data.

Author Response

Thanks for the insightful comments as well as suggestions to improve the paper. We are happy to incorporate them into this revised manuscript.

Your kind decision on this paper will be very much appreciated.

Sincerely,

Chao-Hung Lin
